# *Acinetobacter baumannii* Infection-Related Mortality in Hospitalized Patients: Risk Factors and Potential Targets for Clinical and Antimicrobial Stewardship Interventions

**DOI:** 10.3390/antibiotics11081086

**Published:** 2022-08-10

**Authors:** Diaa Alrahmany, Ahmed F. Omar, Aisha Alreesi, Gehan Harb, Islam M. Ghazi

**Affiliations:** 1Pharmaceutical Care Department, Directorate General of Medical Supplies, Ministry of Health, Muscat 3110, Oman; 2General Medicine Department, Suhar Hospital, Suhar 8484, Oman; 3Pharmacy Department, Suhar Hospital, Suhar 8484, Oman; 4GH Statistics, Cairo 11357, Egypt; 5Arnold and Marie Schwartz College of Pharmacy, Long Island University, Brooklyn, NY 11201, USA

**Keywords:** *Acinetobacter baumannii*, risk factors, mortality, comorbidity, hospitalization, inpatients, MDR, critical care, history of hospitalization, history of infection, bloodstream infection, concurrent infections, antimicrobial stewardship

## Abstract

Due to resistance and scarcity of treatment options, nosocomial *Acinetobacter baumannii* infections are associated with significant fatality rates. We investigated the factors contributing to infection-related deaths to develop tailored stewardship interventions that could reduce these high mortality rates. We reviewed the medical records of adult inpatients with *A. baumannii* infections over two years. Patient demographics and clinical data were collected and statistically analyzed. The study included 321 patients with positive *A. baumannii* microbiological cultures, with respiratory infections accounting for 58.6%, soft tissues 29.3%, bacteremia 8.6%, urine 2.1%, and others 1.4%. The study population’s median (IQR) age was 62.6 (38.9–94.9) years, and hospital stay was 20 (9.5–40) days. Statistical analysis revealed that various risk factors contribute significantly to high in-hospital all-cause mortality (44%), as well as 14-day and 28-day mortality rates. Deaths increased by a factor of 1.04 with every additional year of age (*p* = 0.000), admission to the critical care unit (*p* = 0.000, OR: 2.86), and patients admitted with an infectious diagnosis had nearly three times the mortality rate as those admitted with other diagnoses (*p* = 0.000, OR: 3.12). Male gender (*p* < 0.001, OR: 2.14), any comorbid conditions (*p* = 0.000, OR: 5.29), prolonged hospitalization (>7 days) (*p* = 0.023, OR: 1.98), and hospital acquisition of infection (*p* = 0.027, OR: 1.68) were among the most significant predictors of mortality. All variables were investigated for their impact on all-cause, 14-day, and 28-day mortality rates. Improving multidisciplinary infection control practices, regular disinfection of patient care equipment, and optimal intubation practice that avoids unnecessary intubation are necessary interventions to reduce infection-related mortality rates. Better antibiotic selection and de-escalation, shorter hospital stays whenever possible, prompt medical stabilization of comorbid conditions, and fewer unnecessary admissions to critical care units will all lead to improved outcomes.

## 1. Introduction

Over the last two decades, *Acinetobacter baumannii* has been shown to develop a mounting resistance to nearly all available antibiotics. *A. baumannii* harbors the largest known pathogenicity island, consisting of 45 resistance genes, which enables horizontal gene transfer (HGT) via transposon migration and uptake, disseminating acquired antibiotic resistance to a wide range of antibiotics [1]. *A. baumannii’s* ability to modify its outer membrane, upregulate efflux pumps, form biofilms, colonize medical devices, and acquire resistance determinants explains how it can cause difficult-to-treat and life-threatening infections [2].

Due to the significant increase in infection-related mortality rates due to limited antimicrobial options [3,4,5], the World Health Organization (WHO) designated carbapenem-resistant *A. baumannii* (CRAB) as a high-priority target for antibiotic research and development in 2017. [6]. Similarly, it was classified as a problematic pathogen by the Infectious Diseases Society of America (IDSA) [7]. Furthermore, according to the 2013 Centers for Disease Control and Prevention (CDC) report, multi-drug resistant (MDR) *A. baumannii* is a serious threat that causes approximately 7000 infections and 500 deaths each year in the United States. [8]

*A. baumannii* is a significant contributor to the increased rates of complicated hospital-acquired infections (HAI). According to the National Healthcare Safety Network (2008), *A. baumannii* is responsible for nearly 2% of all catheter-related bloodstream infections (BSIs) and 8% of ventilator-associated pneumonia (VAPs), with mortality rates ranging from 13 to 30% [9]. Many studies have linked chronic comorbid conditions, bedridden status, venous catheterization, intensive care unit (ICU) stay, infections with MDR phenotypes, concurrent fungal infection, and age to *A. baumannii* infection-related mortality [10,11,12].

Previous research by our group [13] found that patients infected with *A. baumannii* had high mortality and treatment failure rates. This study aims to identify risk factors contributing to poor clinical outcomes and higher mortality rates to provide evidence-based risk-mitigation recommendations for our national antimicrobial stewardship program.

## 2. Methods

### 2.1. Study Population

We reviewed patients’ electronic medical records using the following inclusion criteria: adult inpatients (≥18 years) admitted to Suhar hospital (a referral hospital serving a third of national population) with clinical signs and symptoms of infection (to rule out colonization) and laboratory-confirmed identification of *A. baumannii* in infection-site samples. Only the first infection episode in two years (1 January 2018–31 December 2019) was chosen for patients with more than one positive culture. We examined the patients’ demographics, presenting symptoms, underlying comorbid conditions, prior hospitalization, and infection history.

We studied the hospitalization details, including admission diagnosis, discharge status, length of stay, admission ward, infection acquisition location, readmission rates, and microbiological information, including specimen type, susceptibility pattern, resistance phenotype, and concurrent infections. Patients with positive *A. baumannii* cultures who were not admitted or died prior to receiving one dose of antibiotics and pediatric patients were excluded from the study. The primary endpoints were all-cause in-hospital mortality, 14-day, and 28-day mortality before infection signs were resolved.

### 2.2. Phenotype Categorization

Following the CLSI 2010 M100-S20 guidelines, all *A. baumannii* isolates resistant to at least one antibiotic from three or more antimicrobial classes were classified as MDR [14]. Carbapenemases producing isolates were detected phenotypically according to CLSI. Isolates with inhibition zones of <23 mm around (ertapenem 10 g or meropenem 10 g) and resistance to one or more third-generation cephalosporin (e.g., cefotaxime, ceftazidime, and ceftriaxone) were considered as carbapenem-resistant *A. baumannii* (CRAB). All penicillins, cephalosporins, carbapenems, and aztreonam were reported to be resistant in confirmed carbapenemase producing strains.

### 2.3. Definitions

Chronic kidney disease is defined as an estimated glomerular filtration rate (eGFR) of <60 mL/min/1.73 m^2^. Chronic respiratory diseases include chronic obstructive pulmonary disease, bronchiectasis, cystic fibrosis, and asthma. Cerebrovascular diseases include intracranial stenosis, aneurysms, vascular malformations, thrombotic strokes, blockage (embolism), or hemorrhagic strokes. Chronic cardiac diseases include heart failure, hypertension, rheumatic heart disease, cardiomyopathy, arrhythmias, congenital heart disease, valvular heart disease, aortic aneurysms, peripheral artery disease, thromboembolic disease, and venous thrombosis. Immunocompromised patients are those receiving T-cell immunosuppressants, tumor necrosis factor (TNF) blockers, specific monoclonal antibodies, or corticosteroids at a dose of 0.3 mg/kg/d of prednisone for at least 72 h before the index infection. A critical care stay is considered if the patient is admitted to the Intensive Care Unit (ICU), Cardiac Care Unit (CCU), or Burn Unit (BU) for more than 24 h. Death was considered if the symptomatic patient had a positive *A. baumannii* culture and died before the resolution of signs of *A. baumannii* infection during the same hospitalization. The following criteria demonstrate infection resolution: subsiding of presenting symptoms, normal laboratory values of white blood cells (WBC) and C-reactive protein (CRP), or negative culture of the same sample source as the original index infection.

### 2.4. Statistical Analysis

Descriptive statistics were used to summarize demographics and patient characteristics. The χ^2^ test or Fisher’s exact test was used for categorical data, and the Student’s t-test or Wilcoxon rank-sum test was used for numerical data. 

A correlation analysis was performed to investigate the nature and strength of the relationship between risk factors and mortality; *p* values < 0.05 were considered statistically significant. R software (https://www.r-project.org/) was used for all statistical analyses.

## 3. Results

Clinical and microbiological data from 321 symptomatic adult inpatients (total patient days 14108) with confirmed *A. baumannii* clinical infections were reviewed as part of an initiative to evaluate variables associated with all-cause (44%), 14-day (12%), and 28-day (13%) infection-related mortality in the context of evidence-based clinical and antimicrobial recommendations. The cumulative number of risk factors was a significant predictor of mortality; every additional risk factor increased all-cause in-hospital mortality rates by a factor of 1.5 (*p* < 0.05, OR: 1.5). The characteristics of the patients, hospitalization and infection-related details are shown in Table 1, Table 2, and Table 3, respectively.

Hospitalization, patient, and infection-related variables were examined for statistical significance.

### 3.1. Patient-Related Risk Factors

Age: The overall study sample’s median (IQR) age was 62.7 (38.9–75.2); for recovered patients, it was 48 (30.9–70.4), and 69.3 (60.6–76.9) for non-survivals. Patients > 65 years had a higher all-cause mortality rate compared to those ≤ 65 years (*p* = 0.000, OR: 3.41). Age maintains the same effect on 14-day and 28-day mortalities (*p* = 0.040, OR: 2.09) and (*p* = 0.044, OR: 1.99), respectively.

Gender: Both sexes were nearly equally represented in the study sample, with 180 (56%) males and 141 (44%) females. High odds ratios of deaths were noticed in males among all-cause and 28-day mortalities (*p* = 0.001, OR: 2.14) and (*p* = 0.009, OR: 2.73), respectively.

Comorbid conditions: Compared to patients without comorbidities, the presence of any comorbid condition increased all-cause and 14-day mortality rates by nearly five times (*p* = 0.000, OR: 5.29) and (*p* = 0.018, OR: 4.29), respectively. Among all chronic diseases, diabetes mellitus (*p* = 0.000, OR: 2.38), chronic cardiac (*p* = 0.000, OR: 2.72), and respiratory (*p* = 0.007, OR: 2.77) diseases were the most significant predictors of all-cause mortality.

### 3.2. Hospitalization-Related Risk Factors

Length of stay (LOS): The overall study sample’s median (IQR) LOS was 20 (9.5–40) days; for recovered patients, it was 17 (8–35), and 23 (13–50) for non-survivals. Prolonged hospitalization was found to be a significant predictor for 14-day (*p* < 0.000, OR: 0.86) and 28-day (*p* < 0.024, OR: 0.98) mortalities.

Admission to critical care units: Patients admitted to critical care units have a nearly three-fold all-cause mortality rate compared to those admitted to general wards (*p* =0.000, OR: 2.86). Meanwhile, it has no statistically significant effect on 14-day and 28-day mortalities.

History of hospitalization: Both a 90-day history of hospitalization and the hospitalization due to chronic disease management were significant predictors of earlier deaths (14-day) (*p* = 0.024, OR: 2.22) and 28-day (*p* = 0.046, OR: 1.62).

Antibiotic-related adverse events: ADE were found to be significantly associated with all-cause mortality (*p* = 0.000, OR: 3.13) and linked to a non-significant high odds ratio of 28-day deaths.

### 3.3. Infection-Related Risk Factors

Admission diagnosis: When compared to patients admitted with a non-infectious diagnosis, patients with an infectious disease diagnosis had significantly higher rates of all-cause (*p* = 0.000, OR: 3.12), 14-day (*p* < 0.006, OR: 2.98), and 28-day (*p* < 0.01, OR: 2.61) mortality.

Site of infection: Patients with pneumonia had significantly higher all-cause in-hospital mortality rates (*p* = 0.000, OR: 2.85) than other types of infections, while bacteremic patients died earlier (28-day) (*p* = 0.042, OR: 2.83).

MDR-related infections: Patients infected with MDRAB had a significantly higher all-cause mortality rate than those infected with sensitive strains (*p* = 0.000, OR: 3.43). However, this effect completely vanished with the near-term deaths.

Hospital-acquired infections: Hospital-acquired *A. baumannii* infections significantly increase all-cause in-hospital mortality (*p* = 0.027, OR: 1.68) but do not affect other mortalities.

Exposure to infections: A 90-day previous exposure to any bacterial infection (*p* < 0.050, OR: 2.05), particularly Gram-negative bacteria (*p* = 0.026, OR: 2.31) (most notably *P. aeruginosa* (*p* = 0.039, OR: 2.65) and *E. coli* (*p* = 0.039, OR: 2.86), was associated with highly significant odd ratios of early-onset (14-day) deaths.

Concurrent polymicrobial infections: When compared to pure *A. baumannii* infection, late-onset (all-cause) mortality rates were significantly higher in the presence of concurrent infections (*p* = 0.000, OR: 2.66), most likely in concurrent infection with *K. pneumonia* (*p* = 0.001, OR: 2.18), *P. aeruginosa* (*p* = 0.022, OR: 1.70), and other Gram-negative isolates (*p* = 0.000, OR: 2.57).

## 4. Discussion

This research examined 17 variables that were thought to contribute to *A. baumannii* infection-related mortality in hospitalized patients. Binary logistic regression analyses were used. Our findings are discussed as modifiable and non-modifiable risk factors to guide targeted clinical interventions to reduce infection-related mortality in the early and late onsets.

### 4.1. Modifiable Risk Factors 

Prolonged hospitalization session was a significant predictor of mortality rates; patients admitted for >7 days were more likely to die than those admitted for less than 7 days (*p* < 0.05). Among the 259 patients who stayed for > 7 days, 169 (65%) acquired MDRAB infection, significantly contributing to early-onset mortality. While Thorne was unable to link LOS to *A. baumannii*-related acquisition or mortality [15], others have linked LOS to infections and mortality caused by a variety of resistant pathogens such as (methicillin-resistant *S. aureus*, extended-spectrum β-lactamase-producing *K. pneumoniae*, and carbapenem-resistant *A. baumannii*) [16]. Thus, early discharge and home treatment may reduce infection acquisition and treatment-related costs. The antimicrobial stewardship team can promote early discharge strategies by reviewing patient isolation protocols, optimizing surgical prophylaxis protocols, and switching IV to PO whenever possible [17].

Admission to critical care units (92/321) was associated with a non-significant high odds ratio of early-onset mortality while significantly contributing to a higher all-cause mortality rate than general wards (*p* < 0.05). This may be due to excessive catheterization or lax infection control measures in these areas, which may have resulted in its closure to allow disinfection [18]. Many studies support our findings; Choe and colleagues discovered that prior ICU admission significantly contributed to 30-day mortality due to *A. baumannii* (*p =* 0.04) [19]. Meanwhile, Niu and colleagues discovered that ICU stay is a risk factor for 28-day mortality (*p* = 0.04). In an American study, Blanco discovered that patients positive for *A. baumannii* at ICU admission were 15.2 times more likely to develop a subsequent positive clinical culture for *A. baumannii* and 1.4 times more likely to die during the current hospitalization (*p* = 0.01) [20]. In a prospective surveillance study done by Ganesan and colleagues, CRAB ranks second after *K. pneumoniae* as one of the main causative pathogens for device-associated hospital-acquired infections in patients admitted to critical care areas [21].

Intuitively, patients admitted to critical care areas usually suffer a diminished health status, weakened immunity, and are more vulnerable to invasive maneuvers and highly virulent pathogens, necessitating regular disinfection of the ICU environment and equipment, as well as optimal intubation practice with the goal of avoiding excessive catheterization [22]. 

Hospital-acquired infections (HAIs) significantly contribute to all-cause mortality with a non-significant high odds ratio of 28-day mortality. Similarly, hospitals in the United States and Thailand noticed that mortality rates for patients with bloodstream infections (BSIs) caused by community-acquired *A. baumannii* were significantly lower than those for patients with hospital-acquired *A. baumannii* [23,24]. This could be explained by the higher prevalence of virulent phenotypes of *A. baumannii* in hospitals due to increased antimicrobial treatment exposure. Adopting better multidisciplinary infection control practices, regular disinfection of patient care equipment, reducing unnecessary admissions through updated outpatient practices, and increasing outpatient care resources contribute to reduced HAIs. [22].

Concurrent polymicrobial infections were a statistically significant predictor of late-onset mortality (*p* < 0.05), most likely in patients infected with *Klebsiella pneumonia, Pseudomonas aeruginosa,* and other Gram-negative isolates. The ability of *K. pneumoniae*, *P. aeruginosa*, and other Gram-negative isolates to disseminate resistance determinants via horizontal gene transfer (HGT) to *A. baumannii* strains may explain the high prevalence of MDRAB infections in patients with polymicrobial infections (210/239, 88%). Yung and colleagues did not replicate this finding; they found no significant difference in 14-day mortality among patients with monomicrobial and polymicrobial *A. baumannii* BSIs (26.9% vs. 29.2%, *p* < 0.77), with relatively higher rates observed in polymicrobial BSIs with concomitant isolation of *E. coli*, *P. aeruginosa*, and *Enterobacter* spp. [25]. Better identification and monitoring of HGT pathways, robust infection control practices, and regular disinfection of healthcare areas contribute to the limited intra-species spread of resistance determinants [26].

Antibiotic-related adverse events (ADE) were found to be significantly associated with all-cause mortality (*p* < 0.000) and to have a non-significantly high odds ratio of 28-day deaths. Patients (147/321, 46%) experienced adverse events due to prolonged antibiotic use; more likely, 92% fungal infection, 6% acute renal failure, and 2% other adverse events. Moreover, ADR development was associated with a longer LOS median (IQR) of 26 (15–71) compared to 16 (7–31), (*p* < 0.05), in patients who did not develop ADR. Dose and duration optimization, de-escalation of antimicrobial treatment, and therapeutic drug monitoring remain effective tactics to minimize adverse drug reactions [27,28].

### 4.2. Non-Modifiable Risk Factors 

Age maintains a significant directly proportional effect on early-onset and late-onset mortality; patients >65 years had higher mortality rates compared to those ≤65 years, which coincides with a Chinese study that concluded hospital-acquired *A. baumannii* infections in patients aged >65 years were found to be a high-risk predictor of mortality [29].

Males showed statistically significant high odd ratios of all-cause and 28-day mortalities, which is consistent with a multicentered study by many Serbian hospitals [30]. In this study, the correlation of mortality rates with males can be explained by several statistically associated variables, such as the need for critical care unit admission (*p* = 0.02), the acquisition of MDR-related infections (*p =* 0.001), and exposure to polymicrobial infections (*p* = 0.000).

Underlying chronic comorbidities were significant risk factors for mortality (*p* <0.000). Our findings are consistent with those of Chang and colleagues, who identified illness severity, duration of mechanical ventilation, prior hospitalization, and underlying conditions as predictors of mortality [31]. It also matches the finding of a recent (2019) Chinese meta-analysis that linked comorbidities to mortality caused by *A. baumannii* infections [32]. Others concluded that underlying illnesses are more influential than the infection itself as a cause of death [33], implying that concurrent comorbidities to *A. baumannii* infections must be clinically stabilized as soon as possible to improve outcomes. Among all chronic diseases, diabetes mellitus (*p* < 0.05), chronic cardiac (*p* < 0.05), and respiratory (*p* < 0.05) diseases were the most significant predictors of all-cause mortality.

MDR-related infections were associated with a significantly higher all-cause mortality rate than those infected with sensitive strains (*p* < 0.05, OR: 3.43). However, it was still associated with non-significant high odds ratio of near-term deaths. Patients with MDRAB infections had a high incidence of polymicrobial infections (88%) and chronic comorbidities (66%); these were likely the confounders contributing to increased mortality. Our finding matches many studies from different parts of the world; they found several folds increase in mortality rate in MDRAB infected patients than in those infected with susceptible phenotypes, as well as increased hospital and intensive care unit length of stay [34,35,36].

Patients who presented on admission with an infectious disease diagnosis had significantly higher mortality rates of all types. Patients with pneumonia, in particular, had the highest mortality rates of any infection; meanwhile, bacteremic patients died earlier. The high mortality rates among respiratory and BSIs can be explained by excessive intubation and catheterization, particularly in critical care areas, which may serve as a reservoir for resistant phenotypes; the high incidence of MDRAB infections among both groups (respiratory and bacteremia patients); (85%) and (82%), respectively explains mounted deaths. A high death rate was reported in France due to case severity and limited antibiotic options against MDRAB respiratory and bloodstream infections. [37].

History of hospitalization—primarily due to a chronic condition—and exposure to infection were highly significant predictors of early-onset mortality. Exposure to bacterial infection, particularly *P. aeruginosa* and *E. coli*, were associated with significantly high odds of early-onset (14-day) deaths. Many studies linked morbidity and mortality risks to previous healthcare exposure [38,39], which is an alarming sign to adopt a firm infection control policy. 

History of infection with *P. aeruginosa* and *E. coli* contributed significantly to early onset mortality which may be due to disseminated resistance determinants to current isolates. Further studies are needed to assess the impact of prior exposure to certain bacterial strains that may serve as a source of transferrable resistance determinants and the exposure to potentially high resistance-inducing antimicrobials. Identifying these factors may help identify patients at high risk for MDRAB infection-related mortality and place extra emphasis on early intervention. It would also be conducive in any endeavors aiming to construct an algorithm for early detection and prompt treatment of high-risk patient groups.

While we incorporated numerous possible risk factors in our analysis, a limitation to our study could be that the characteristics of the cohort studied might not be generalizable to other geographic locations, and consideration of our findings in conjunction with other similar studies could consolidate the construction of a robust model to predict and mitigate risk factors for mortality.

## 5. Conclusions

Old age, prolonged hospitalization, inadequate infection control, resorting to invasive procedures, host health status, and antibiotic overuse appear to be the most important risk factors for nosocomial infection-related mortality. Nosocomial infections are avoidable if we improve the operation of healthcare services based on multidisciplinary infection control practice, regular disinfection of patient care equipment, and bedside antiseptic dispensers with instructions to remind staff to clean their hands.

Improved outcomes may result from better intubation practice, shorter hospital stays, prompt medical stabilization of comorbidities, and limiting unnecessary admissions to critical care areas. The antimicrobial stewardship team can promote early discharge and minimize HAI exposure through optimal antibiotic selection and de-escalation, optimized surgical prophylaxis protocols, and IV to PO switching tactics whenever possible. More research is needed to investigate the effect of antibiotics on the dynamics of *A. baumannii* microbial resistance in relation to multidrug-resistant outbreaks.

## Figures and Tables

**Table 1 antibiotics-11-01086-t001:** Patient-related variables versus mortality rates (Binary logistic regression).

	All-Cause in-Hospital Mortality *n* = 140 (44%)					14-Day Mortality *n* = 37 (12%)					28-Day Mortality *n* = 41 (13%)				
	No	%	*p*	OR	CI	No	%	*p*	OR	CI	No	%	*p*	OR	CI
Age, median (IQR)	69	(60–76)	0.000	1.04	1.03–1.06	73	(60–77)	0.006	1.03	1.01–1.05	68	(56–76)	0.020	1.02	1.00–1.04
Age > 65 years	88	62.9%	0.000	3.41	2.15–5.42	23	62.2%	0.040	2.09	1.03–4.23	25	61.0%	0.044	1.99	1.02–3.90
Male	93	66.4%	0.001	2.14	1.35–3.37	24	64.9%	0.254	1.51	0.74–3.09	31	75.6%	0.009	2.73	1.29–5.77
Female	47	33.6%		0.47	0.30–0.74	13	35.1%		0.66	0.32–1.35	10	24.4%	0.009	0.37	0.17–0.78
Any comorbidities	126	90.0%	0.000	5.29	2.82–9.92	34	91.9%	0.018	4.29	1.28–14.37	35	85.4%	0.101	2.13	0.86–5.28
Diabetes	74	52.9%	0.000	2.38	1.51–3.75	19	51.4%	0.182	1.60	0.80–3.18	22	53.7%	0.084	1.79	0.93–3.46
Chronic renal failure	32	22.9%	0.070	1.69	0.96–2.98	10	27.0%	0.153	1.78	0.81–3.91	8	19.5%	0.841	1.09	0.47–2.50
Active malignancy	7	5.0%	0.623	1.31	0.45–3.82	2	5.4%	0.742	1.30	0.28–6.03	1	2.4%	0.526	0.51	0.07–4.03
Immuno-suppressed	4	2.9%	0.959	1.04	0.27–3.93	1	2.7%	0.968	0.96	0.12–7.89	0	0.0%	*	*	*
Chronic cardiac diseases	97	69.3%	0.000	2.72	1.71–4.33	28	75.7%	0.012	2.74	1.25–6.02	27	65.9%	0.166	1.62	0.82–3.23
HIV follow-up/AIDS	1	0.7%	0.855	1.30	0.08–20.89	1	2.7%	0.148	7.86	0.48–128.43	0	0.0%	*	*	*
Chronic respiratory disease	23	16.4%	0.007	2.77	1.33–5.78	6	16.2%	0.275	1.70	0.66–4.42	9	22.0%	0.019	2.75	1.18–6.38

No.: number, *p*: probability value, OR: odds ratio, CI: confidence interval, IQR: intra-quartile range, *: value cannot be produced by software.

**Table 2 antibiotics-11-01086-t002:** Hospitalization details versus mortality rates (Binary logistic regression).

	All-Cause in-Hospital Mortality *n* = 140 (44%)					14-Day Mortality *n* = 37 (12%)					28-Day Mortality n = 41 (13%)				
	No.	%	*p*	OR	CI	No.	%	*p*	OR	CI	No.	%	*p*	OR	CI
LOS, median (IQR)	23	(13–50)	0.193	1.00	1.00–1.00	7	(5–11)	0.000	0.86	0.81–0.92	18	(16–22)	0.024	0.98	0.96–1.00
LOS > 7 days	121	86.4%	0.023	1.98	1.10–3.59	18	48.6%	0.000	0.17	0.08–0.35	41	100.0%	*	*	*
Referred from another hospital	65	46.4%	0.997	1.00	0.64–1.56	17	45.9%	0.951	0.98	0.49–1.95	17	41.5%	0.496	0.79	0.41–1.54
Admitted from community	75	53.6%		1.00	0.64–1.56	20	54.1%		1.02	0.51–2.03	24	58.5%		1.26	0.65–2.45
Admission to critical care area	57	40.7%	0.000	2.86	1.74–4.72	13	35.1%	0.356	1.41	0.68–2.90	14	34.1%	0.407	1.34	0.67–2.69
Admission to general ward	83	59.3%		0.35	0.21–0.58	24	64.9%		0.71	0.35–1.47	27	65.9%		0.74	0.37–1.49
Admission with ID diagnosis	97	69.3%	0.000	3.12	1.96–4.96	28	75.7%	0.006	2.98	1.36–6.55	30	73.2%	0.010	2.61	1.26–5.42
Admission with a non-ID diagnosis	43	30.7%		0.32	0.20–0.51	9	24.3%		0.34	0.15–0.74	11	26.8%		0.38	0.18–0.79
6 months H/O invasive procedure	55	39.3%	0.280	0.78	0.50–1.22	12	32.4%	0.184	0.61	0.30–1.26	12	29.3%	0.067	0.51	0.25–1.05
90-day H/O hospitalization	45	32.1%	0.985	1.00	0.63–1.61	18	48.6%	0.024	2.22	1.11–4.43	14	34.1%	0.762	1.11	0.56–2.22
Hospitalization due to chronic illness	26	18.6%	0.096	2.41	1.09–5.35	10	27.0%	0.046	1.62	0.58–4.51	12	29.3%	0.008	9.26	1.95–43.89
Hospitalization due to acute illness	19	13.6%	0.096	0.41	0.19–0.92	8	21.6%	0.046	0.62	0.22–1.72	2	4.9%	0.008	0.11	0.02–0.51

No.: number, *p*: probability value, OR: odds ratio, CI: confidence interval, IQR: intra-quartile range, ID: infectious disease, LOS: length of stay, H/O: history of, *: value cannot be produced by software.

**Table 3 antibiotics-11-01086-t003:** Infection-related details versus mortality rates (Binary logistic regression).

	All-Cause in-Hospital Mortality *n* = 140 (44%)					14-Day Mortality *n* = 37 (12%)					28-Day Mortality n = 41 (13%)				
	No.	%	*p*	OR	CI	No.	%	*p*	OR	CI	No.	%	*p*	OR	CI
Blood sample	12	8.6%	0.288	1.60	0.67–3.83	4	10.8%	0.317	1.79	0.57–5.61	6	14.6%	0.042	2.83	1.04–7.71
Patient-related device sample	2	1.4%	0.207	0.36	0.07–1.76	0	0.0%	*	*	*	1	2.4%	0.880	0.85	0.10–6.98
Respiratory sample	82	58.6%	0.000	2.85	1.81–4.50	19	51.4%	0.356	1.38	0.70–2.74	20	48.8%	0.531	1.23	0.64–2.38
Skin & soft tissue sample	41	29.3%	0.211	0.74	0.46–1.19	14	37.8%	0.509	1.27	0.63–2.58	14	34.1%	0.870	1.06	0.53–2.12
Urine sample	3	2.1%	0.000	0.08	0.02–0.26	0	0.0%	*	*	*	0	0.0%	*	*	*
Infection with MDRAB	127	90.7%	0.000	3.43	1.77–6.63	29	78.4%	0.627	0.81	0.35–1.88	37	90.2%	0.125	2.31	0.79–6.76
Infection with sensitive AB	13	9.3%		0.29	0.15–0.56	8	21.6%		1.23	0.53–2.85	4	9.8%		0.43	0.15–1.26
Hospital-acquired infection	93	66.4%	0.027	1.68	1.06–2.65	9	24.3%	0.000	0.18	0.08–0.40	29	70.7%	0.120	1.76	0.86–3.59
Community-acquired infection	47	33.6%		0.60	0.38–0.94	28	75.7%		5.55	2.52–12.22	12	29.3%		0.57	0.28–1.16
90-day recurrence of infection	10	7.1%	0.082	0.50	0.23–1.09	0	0.0%	*	*	*	0	0.0%	*	*	*
H/O exposure to Gram-negative	29	20.7%	0.951	0.98	0.57–1.69	13	35.1%	0.026	2.31	1.10–4.82	9	22.0%	0.856	1.08	0.49–2.38
Any H/O exposure to infection	33	23.6%	0.704	0.91	0.54–1.51	14	37.8%	0.050	2.05	1.00–4.21	10	24.4%	0.972	0.99	0.46–2.12
H/O exposure to *K. pneumoniae*	8	5.7%	0.295	0.63	0.26–1.51	4	10.8%	0.416	1.60	0.52–4.97	4	9.8%	0.554	1.41	0.46–4.34
H/O exposure to *P. aeruginosa*	14	10.0%	0.723	1.15	0.54–2.44	7	18.9%	0.039	2.65	1.05–6.69	5	12.2%	0.504	1.42	0.51–3.94
H/O exposure to *E. coli*	13	9.3%	0.282	1.58	0.69–3.65	6	16.2%	0.039	2.86	1.06–7.74	3	7.3%	0.967	0.97	0.28–3.42
H/O exposure to Gram-positive	15	10.7%	0.950	1.02	0.50–2.09	6	16.2%	0.243	1.77	0.68–4.61	6	14.6%	0.371	1.54	0.60–3.99
H/O exposure to MRSA	4	2.9%	0.468	0.64	0.19–2.16	2	5.4%	0.573	1.57	0.33–7.44	1	2.4%	0.642	0.61	0.08–4.86
H/O exposure to MSSA	4	2.9%	0.468	0.64	0.19–2.16	0	0.0%	*	*	*	3	7.3%	0.209	2.38	0.62–9.17
Polymicrobial infection	118	84.3%	0.000	2.66	1.53–4.61	21	56.8%	0.010	0.40	0.20–0.81	35	85.4%	0.093	2.17	0.88–5.37
Concurrent infection with Gram-negative	107	76.4%	0.000	2.57	1.58–4.18	16	43.2%	0.005	0.37	0.18–0.73	30	73.2%	0.232	1.56	0.75–3.25
Concurrent infection *K. pneumoniae*	70	50.0%	0.001	2.18	1.38–3.43	6	16.2%	0.004	0.26	0.11–0.64	20	48.8%	0.199	1.54	0.80–2.97
Concurrent infection *P. aeruginosa*	64	45.7%	0.022	1.70	1.08–2.67	9	24.3%	0.062	0.47	0.21–1.04	12	29.3%	0.190	0.621	0.30–1.27
Concurrent infection *E. coli*	29	20.7%	0.343	1.32	0.75–2.32	4	10.8%	0.214	0.50	0.17–1.48	7	17.1%	0.817	0.903	0.38–2.15
Concurrent infection with Gram-positive	69	49.3%	0.016	1.73	1.11–2.72	11	29.7%	0.119	0.55	0.26–1.16	17	41.5%	0.969	0.99	0.51–1.92
Concurrent infection MRSA	20	14.3%	0.791	1.09	0.58–2.07	2	5.4%	0.136	0.33	0.08–1.42	6	14.6%	0.853	1.09	0.43–2.77
Concurrent infection MSSA	8	5.7%	0.060	1.31	0.48–3.58	3	8.1%	0.360	1.84	0.50–6.78	1	2.4%	0.435	0.44	0.06–3.44
Antibiotic-related ADE	86	61.4%	0.000	3.13	1.98–4.96	16	43.2%	0.741	0.89	0.45–1.78	24	58.5%	0.082	1.80	0.93–3.50

No.: number, *p*: probability value, OR: odds ratio, CI: confidence interval, IQR: intra-quartile range, ID: infectious disease, LOS: length of stay, H/O: history of, MDRAB: multi-drug resistant *A. baumannii*, AB: *A. baumannii*, MRSA: methicillin-resistant *Staphylococcus aureus*, MSSA: Methicillin-sensitive *Staphylococcus aureus*, ADE: adverse drug event, *: value cannot be produced by the software.

## Data Availability

Data supporting results reported in this manuscript are available at Suhar Hospital upon request and approval.

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
