# Peer review of "Acinetobacter baumannii Infection-Related Mortality in Hospitalized Patients: Risk Factors and Potential Targets for Clinical and Antimicrobial Stewardship Interventions"

_antibiotics, 2022, doi:10.3390/antibiotics11081086_

Round 1
Reviewer 1 Report
Alrahmany and colleagues reported for the development of antimicrobial stewardship interventions to reduce mortality of nosocomial A. baumannii infections.
Major comments are suggested for:
Line 17, mention the source of culture, for examples, “patients with positive blood cultures for A.baumannii” or “patients diagnosed with A. baumannii infections in different body fluids”. How did A. baumannii infections cause high all-cause mortality (44%), for example, in the lung?
Line 23, Could you provide any specific reason for the result of gender difference especially in 28-day mortality (Table 1), for examples, antibiotic resistance or other contributing factors based on the medical records?
Tables, It is hard to compare the results. Divide the columns of categories and criteria. Give the exact P-values instead of <0.05 in tables and text. What indicates the colors of the cells, yellow, P < 0.1 (in Table 3), orange P < 0.05, green P < 0.01?
Titles of tables should be clear, for example, “Table 1. Chi-square test results of subgroups from the comparison of patient demographics”
In Table 2, move “Any Previous exposure to infection to above the “Exposure to Gram-negative bacteria”. Is the “Exposure to Gram-negative bacteria” subdivided by K. pneumoniae, E. coli, and P. aeruginosa? Is this categorization also used in “Exposure to Gram-positive bacteria”, “Concurrent infection with Gram negative bacteria” and “Concurrent infection with Gram positive bacteria”? Then, re-organize the table according to the categories. Why is there a significant difference in the odd ratio of “Concurrent infection with Gram +ve”, without no significant differences in those of MRSA and MSSA? Was there other Gram-positive bacteria contributing to all-cause mortality?
In Statistical analysis, It is unclear how to use χ2 test or Fisher exact test for categorical data, and where are numerical data for Student t-test or Wilcoxon rank-sum test. Where was the cumulative effect of all variables with a P< 0.2 in the bivariate analysis shown in a multiple stepwise logistic regression model? There is no Figure 1 showing CART (classification and regression tree analysis) or a correlation analysis of the relationship between risk factors and mortality.
Minor comments:
Line 29, “such challenges” are “the mortality rates”?
Lines 38-41, The sentences cited in the same reference could be combined, as “A. baumannii harbors a pathogenicity island carrying multiple antibiotic resistance genes, previously reported 45 genes, which enable horizontal gene 39 transfer (HGT) via transposon migration and uptake, disseminating acquired antibiotic resistance to a wide range of antibiotics”
Line 50, multi-drug resistant (MDR)
Lines 84 and 85, delete one repeated sentence.
Line 86, add spaces between 10 and g (gram). Check the dose (g or μg) of antibiotics.
Line 107, list the normal values (ranges) of all laboratory tests in Supplementary Table. Delete the undefined terms (WBC, PCR,.... Etc).
In Table 2, K. Pneumoniae -> K. pneumoniae. All taxonomic names should be italicized.
Gram -ve -> “Gram-negative bacteria”. Gram +ve -> “Gram-positive bacteria
Lines 141 and 142, P = 0.040, P = 0.044.
Line 203, italicize “S. aureus”.
Line 306, Extreme -> “Old”
Author Response
|
Reviewer 1 |
|
Alrahmany and colleagues reported for the development of antimicrobial stewardship interventions to reduce mortality of nosocomial A. baumannii infections. |
|
Line 17, mention the source of culture, for examples, “patients with positive blood cultures for A. baumannii” or “patients diagnosed with A. baumannii infections in different body fluids”. How did A. baumannii infections cause high all-cause mortality (44%), for example, in the lung? |
|
Thanks for your comment; more details are added to the sentence to describe the culture source.
“Over two years, 321 patients with positive A.baumannii microbiological cultures were included, with respiratory infections accounting for 58.6%, soft tissues 29.3%, bacteremia for 8.6%, urine for 2.1%, and others 1.4%.” |
|
|
|
Line 23, Could you provide any specific reason for the result of gender difference especially in 28-day mortality (Table 1), for examples, antibiotic resistance or other contributing factors based on the medical records? |
|
Thanks for your comment; contributing factors are added to the gender in the discussion section.
In this study, the correlation of mortality rates with males can be explained by several statistically associated variables, such as the need for critical care unit admission [P=0.02], the acquisition of MDR-related infections [0.001], and exposure to polymicrobial infections [P=0.000]. |
|
|
|
Tables, It is hard to compare the results. Divide the columns of categories and criteria. Give the exact P-values instead of <0.05 in tables and text. What indicates the colors of the cells, yellow, P < 0.1 (in Table 3), orange P < 0.05, green P < 0.01? |
|
Thank you for the comment § Tables are redesigned to be more clear to allow results comparison. § P-values are changed to the exact values all over the tables and manuscript. § Colors are removed from table cells. |
|
|
|
Titles of tables should be clear, for example, “Table 1. Chi-square test results of subgroups from the comparison of patient demographics” Thank you for the comment
|
|
Table titles are modified to clarify the category of data, and the statistical approach |
|
|
|
In Table 2, move “Any Previous exposure to infection to above the “Exposure to Gram-negative bacteria”. Is the “Exposure to Gram-negative bacteria” subdivided by K. pneumoniae, E. coli, and P. aeruginosa? Is this categorization also used in “Exposure to Gram-positive bacteria”, “Concurrent infection with Gram negative bacteria” and “Concurrent infection with Gram positive bacteria”? Then, reorganize the table according to the categories. Why is there a significant difference in the odd ratio of “Concurrent infection with Gram +ve”, without no significant differences in those of MRSA and MSSA? Was there other Gram-positive bacteria contributing to all-cause mortality? |
|
Thanks for your comment § All tables are reorganized according to category § History of bacterial infection: Gram-negative bacteria predominate in the 90-day infection history of all types of mortalities. On the other hand, Gram-positive infections were scarce compared to Gram-negative infections (very clear from the number of cases in the table). Meanwhile, MSSA and MRSA account for more than 85% of Gram-positive 90-day prior infections in our study, but neither contributes significantly to mortality. § In terms of concurrent infections, Gram-positive showed a weak contribution to all-cause mortality (P=0.016) with a non-statistically significant trend with MSSA (P=0.066). MSSA and MRSA are the most common Gram-positive infections, with CoNS and S. agalactiae causing rare infections. When Gram-positive bacteria were segmented into individual pathogens, statistical significance vanished. |
|
|
|
In Statistical analysis, It is unclear how to use χ2 test or Fisher exact test for categorical data, and where are numerical data for Student t-test or Wilcoxon rank-sum test. Where was the cumulative effect of all variables with a P< 0.2 in the bivariate analysis shown in a multiple stepwise logistic regression model? There is no Figure 1 showing CART (classification and regression tree analysis) or a correlation analysis of the relationship between risk factors and mortality. Thank you for the comment |
|
Many articles of literature criticize the use of multiple stepwise logistic regression in clinical studies; because it may produce clinically illogical results, the entire model is excluded from the results section. |
|
|
|
Line 29, “such challenges” are “the mortality rates”? |
|
Thanks for comment Sentence rephrased |
|
|
|
Lines 38-41, The sentences cited in the same reference could be combined, as “A. baumannii harbors a pathogenicity island carrying multiple antibiotic resistance genes, previously reported 45 genes, which enable horizontal gene 39 transfer (HGT) via transposon migration and uptake, disseminating acquired antibiotic resistance to a wide range of antibiotics” |
|
Thanks for comment Sentence rephrased |
|
|
|
Line 50, multi-drug resistant (MDR) |
|
Corrected |
|
|
|
Lines 84 and 85, delete one repeated sentence. |
|
Thanks for the remark The repeated sentence is deleted |
|
|
|
Line 86, add spaces between 10 and g (gram). Check the dose (g or μg) of antibiotics. |
|
Corrected |
|
|
|
Line 107, list the normal values (ranges) of all laboratory tests in Supplementary Table. Delete the undefined terms (WBC, PCR,.... Etc). |
|
Abbreviations are spelled out |
|
|
|
In Table 2, K. Pneumoniae -> K. pneumoniae. All taxonomic names should be italicized. |
|
All bacterial names are italicized |
|
|
|
Gram -ve -> “Gram-negative bacteria”. Gram +ve -> “Gram-positive bacteria |
|
Corrected |
|
|
|
Lines 141 and 142, P = 0.040, P = 0.044. |
|
Corrected |
|
|
|
Line 203, italicize “S. aureus”. |
|
Corrected |
|
|
|
Line 306, Extreme -> “Old” |
|
Changed |

Reviewer 2 Report
selfreference 13 does not mention mortality, irrelevant citation.
2.1
should mention which population generates the study material, investigation site (one hospital?) time period of inclusion. Does the hospital(s) use prophylactic measures according CDC-NHSN or any other systematic infection cotrol system?
3. results
first line: add total patient days, makes it more comparable with other studies
first line. 17 variables? I find 13 variables in 3.1 - 3.3
3.1, 3.2, 3.3 should use the same categorisation (preferable modifiabale/non modifiable), order and the same wording of the topics as 4.1 4.2
Exposure and site of infection is missing in section 4.)
4.1 prolonged hospitalisation is a well known risk factor.
Sunenshine RH, Multidrug-resistant Acinetobacter infection mortality rate andlength of hospitalization. Emerg Infect Dis. 2007 Jan;13(1):97-103.
5.
discussion of the following paper is warranted.
Ganesan V,
Device-Associated Hospital-Acquired Infections: Does Active Surveillance With Bundle Care Offer a Pathway to Minimize Them? Cureus. 2021 Nov 7;13(11)
Author Response
|
Reviewer 2 |
|
selfreference 13 does not mention mortality, irrelevant citation. |
|
Citation removed |
|
|
|
2.1 should mention which population generates the study material, investigation site (one hospital?) time period of inclusion. Does the hospital(s) use prophylactic measures according CDC-NHSN or any other systematic infection cotrol system? |
|
§ Thanks for the remarks § Hospital details and period of inclusion are added under methods (study population) § The hospital has an infection control team operating under measures imposed Oman government |
|
3. results first line: add total patient days, makes it more comparable with other studies first line. 17 variables? I find 13 variables in 3.1 - 3.3 |
|
Thanks for the comment § Total patient days are added in the results section (14108) § All studied variables are mentioned in the method (an appendix is attached), while, only statistically significant variables are discussed in the results section. |
|
|
|
3.1, 3.2, 3.3 should use the same categorisation (preferable modifiabale/non modifiable), order and the same wording of the topics as 4.1 4.2 |
|
We categorize the variables in the results sector as per the type of variable to ease following up the results in the tables (same categorization) In the discussion, we categorized them as modifiable/non-modifiable to be more relevant to our ultimate goal of suggesting targeted interventions to reduce infection-related mortality. |
|
|
|
Exposure and site of infection is missing in section 4.) |
|
Thanks for the comment, we added it under non-modifiable variables History of infection with P. aeruginosa and E.coli contributed significantly to early onset mortality which may be due to disseminated resistance determinants to current isolates. |
|
|
|
4.1 prolonged hospitalisation is a well known risk factor. Sunenshine RH, Multidrug-resistant Acinetobacter infection mortality rate and length of hospitalization. Emerg Infect Dis. 2007 Jan;13(1):97-103. |
|
This article is already cited (reference no.35) |
|
5. discussion of the following paper is warranted. Ganesan V, Device-Associated Hospital-Acquired Infections: Does Active Surveillance With Bundle Care Offer a Pathway to Minimize Them? Cureus. 2021 Nov 7;13(11) |
|
Thanks for the valuable suggestion Study is discussed and cited under admission to critical care units |
Round 2
Reviewer 1 Report
Reviewer’s comments
Minor comments are suggested for:
In Abstract:
Is this meaning? “This study included a total of 321 patients with A. baumannii infections of respiratory tract (58.6%), soft tissues (29.3%), bacteremia (8.6%), urine (2.1%), and others (1.4%).”
In Methods
in page 2, “… episode in 2 years (January, 1st 2018-December 31st, 2019) was chosen ...” place a space between parenthesis and verb “was”
In page 3, “normal laboratory values of white blood cells (WBC) and C-reactive protein (CRP), or …”
In Table 3 and Discussion, “K. Pneumoniae” -> “K. pneumoniae”. The italicized species name must start with a lowercase letter.
“MRSA” and “MSSA” must not be italicized. In footnotes, distinguish the italicized genus and species names. MRSA: methicillin-resistant Staphylococcus aureus, MSSA: Methicillin-sensitive Staphylococcus aureus
In page 8, 3.2. Hospitalization-related risk factors Length of stay (LOS): “… a significant predictor for 14-day for 14-day [P<0.000, OR: 0.86] and 28-day [P><0.024, OR: 0.98] mortalities><0.000, OR: 0.86] and [P = 0.024 OR: 0.98] mortalities.” Please use the absolute P-values determined in Tables.
In page 11, “History of infection with P. aeruginosa and E. coli …” Italicize the taxonomic names of bacteria.
Author Response
Dear reviewer,
Thank you for your comments. Detailed response is attached.
Best,

Reviewer 2 Report
study with data about a. baumanii with limited clinical utility since the modifiable risk factors are subject to infection control measures which should be implemented in any case
Author Response
Dear reviewer,
Thank you for your comments that served to improve the manuscript.
Best,